# Rapid Effects of BCI-Based Attention Training on Functional Brain Connectivity in Poststroke Patients: A Pilot Resting-State fMRI Study

**Larisa Mayorova** [1,2,*], **Anastasia Kushnir** [1], **Viktoria Sorokina** [2], **Pranil Pradhan** [2,3], **Margarita Radutnaya** [2], **Vasiliy Zhdanov** [2], **Marina Petrova** [2,3] **and Andrey Grechko** [2,3]

1   Institute of Higher Nervous Activity and Neurophysiology of the Russian Academy of Sciences, Laboratory of Physiology of Sensory Systems, 117485 Moscow, Russia
2   Federal Research and Clinical Center of Intensive Care Medicine and Rehabilitology, 107031 Moscow, Russia
3   Department of Anesthesiology and Resuscitation with Medical Rehabilitation Courses, Peoples' Friendship University of Russia (RUDN University), 117198 Moscow, Russia
*   Correspondence: larimayor@gmail.com

**Abstract:** The prevalence of stroke-induced cognitive impairment is high. Effective approaches to the treatment of these cognitive impairments after stroke remain a serious and perhaps underestimated challenge. A BCI-based task-focused training that results in repetitive recruitment of the normal motor or cognitive circuits may strengthen stroke-affected neuronal connectivity, leading to functional improvements. In the present controlled study, we attempted to evaluate the modulation of neuronal circuits under the influence of 10 days of training in a P3-based BCI speller in subacute ischemic stroke patients.

**Keywords:** BCI; neurorehabilitation; cognitive training; stroke; attention; functional connectivity; RS-fMRI; thalamus

## 1. Introduction

According to the WHO, stroke is currently ranked the second leading cause of death in the world and remains one of the leading causes of disability. The prevalence of cognitive impairment caused by stroke is high, up to 59% [1]. Attention and speed are the most impaired domains at <1 month to 3 months [2], according to a standardized neuropsychological battery assessing six cognitive domains.

Effective approaches to restoring these cognitive domains after stroke remain a serious and perhaps underappreciated challenge. To date, effective methods of cognitive restoration have been described for speech impairment and neglect syndrome. However, methods of restoring attention, so often damaged and underlying almost all everyday life and occupational tasks, remain elusive.

One group of techniques that have recently been shown to promote neuronal plasticity, rehabilitative BCI training, can theoretically be applied to improve cognitive functions in the cohort of patients under discussion.

Brain–computer interface (BCI) is currently in consideration of being the rational way for cognitive training based on biofeedback [3–6]. The physiological evidence in some studies has shown that applying BCI, in addition to direct personal training, influenced specific frequencies of the electrical activity of the brain measured by EEG, such as theta, alpha, alpha/theta ratio, beta, gamma, sensorimotor rhythms, and others [7–10]. Biofeedback training is a set of measures to train a person to consciously control one or more physiological parameters. The signals of physiological parameters, recorded by special sensors, are amplified and presented to the subject or patient in the form of an accessible "metaphor". The "metaphor" can be a sound indicating the success or failure of a self-regulatory act,

moving "thermometer" bars or other graphic images, or the development of a game story depending on the values of the target parameter. A variety of signals are used, including: peripheral temperature, respiratory parameters, heart rate, electromyogram, and brain activity (EEG-, fMRI-, or fNIRS).

A common neurofeedback training is an exercise in which the patient repeatedly attempts to consciously change their physiological parameters, guided by the "metaphor" of their condition presented on the screen. This training involves a large number of repetitions and is monotonous and tedious. Some participants go through several trial-and-error attempts over many sessions before they manage to achieve some success in self-regulating their desired brain patterns.

In an ideal situation, the positive effect of training should extend to all functions of the trained domain (or better to adjacent cognitive domains), i.e., it should appear not only in the trained task paradigm but also in the untrained tasks, which in the literature is called training transfer. The existing overlap hypothesis of training transfer [11,12] suggests that successful transfer effects are based, first, on similar functional processes and, second, on the corresponding activation of similar neural circuits (functional and neural overlap). According to this, the weak point of biofeedback training for cognitive recovery is the questionable functional overlap (the metaphor management task, strictly speaking, is quite specific and far from an everyday task). Another problem of biofeedback training is its high dependence on the definition of the target rhythm, the detection of which is difficult in stroke.

In the context of current ideas about functional and neural overlap in training transfer, it may seem more rational to use BCIs that involve the patient in more naturalistic tasks and work based on processing the activity of neural circuits associated with the realization of a number of cognitive functions, such as various types of attention, working memory, etc.

There is already evidence of the effectiveness of training in the 3D game environment of the BCI in correcting attention deficit hyperactivity disorder (ADHD); during the training, children were trained to apply the acquired skills to solve academic problems (solving mathematical and language exercises immediately after each training session). According to a randomized controlled trial, this training improved attention in these patients [4]. It described a method of using BCI to treat cognitive decline in the elderly. In a pilot study, BCI was shown to improve memory and attention function in older subjects [3].

A BCI-based task-focused training that results in repetitive recruitment of the normal motor or cognitive circuits may strengthen stroke-affected neuronal connectivity, leading to functional improvements.

Such BCIs include, in particular, P3-based BCI spellers. The P300 cognitive evoked potential, according to the literature, reflects the functioning of the distributed neural network and is directly linked to the realization of executive functions such as different types of attention, working memory, etc. [13]. Since the P300 is generated easily and naturally in response to targeted stimuli, the patient can effectively interact with the proposed learning tool in the BCI environment.

There is very little evidence of the use of P3-based BCI spellers in poststroke patients, nevertheless reflecting the ability of such patients to work with this type of interface with varying degrees of success [14,15]. Regarding the neuromodulatory impact of this type of BCI on cognitive recovery in patients after stroke, as far as we know, there are little data available.

In the present controlled study, we attempted to evaluate the modulation of neuronal circuits under the influence of 10 days of training in a P3-based BCI speller in subacute ischemic stroke patients.

## 2. Materials and Methods

### 2.1. Participants

All patients selected for the study underwent examination by a neurologist, neuropsychologist, and resting-state fMRI procedure, after which they were randomly assigned to the main group or the delayed rehabilitation/therapy group (control group).

Inclusion criteria for both groups were: duration of stroke less than 3 months, age 22 to 82 years, mild to moderate cognitive impairment, and executive function deficits. Exclusion criteria included severe aphasia, epileptic seizures, neuroinfection, claustrophobia, and the presence of a pacemaker or other contraindications for MRI. Conditions such as severe affective disorder were also excluded from the study group to make it as homogeneous as possible in terms of rehabilitative potential.

Patients in the main group then underwent a 10-day BCI training course, while patients in the control group, before randomization, underwent only one first session to assess their ability to work in the BCI environment. At the end of the training course (10 days for patients in the control group), all patients were re-examined by a neurologist, neuropsychologist, and psychiatrist and underwent a resting-state fMRI procedure. All the studies were conducted immediately before the first BCI session and immediately after the last session. Thus, the time period between the two tests was strictly limited to the time of training using the BCI speller. The flowchart of patient selection is presented in Figure 1.

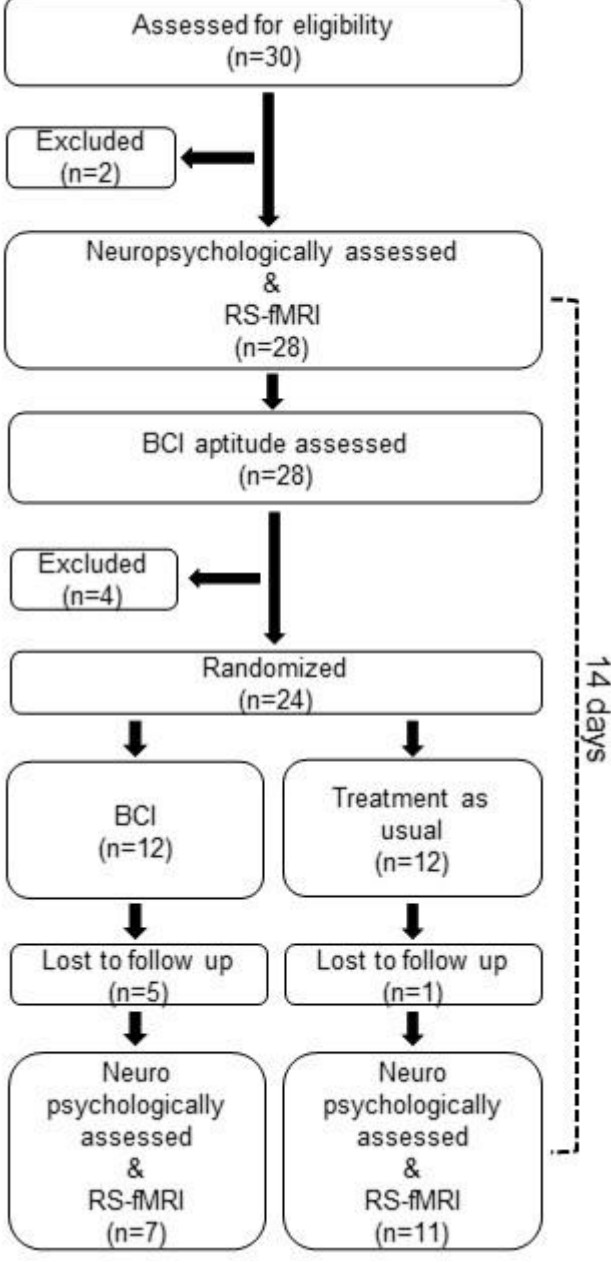

**Figure 1.** Flowchart of patient selection.

As part of the standard comprehensive rehabilitation therapy, patients were treated with: physical therapy, massage, paretic limb position therapy, mechanotherapy, and cardiorespiratory training.

There were 18 participants in the study (7 in the main group and 11 in the control group). The mean age was 62 (interquartile range (IQR) = 10.5) years for the main group and 60 (19.5) years for the control group. Duration of disease was 1.9 (0.48) for the main group and 1.1 (1.48) for the control group. The lesion size was 2.14 (17.08) for the main group and 1.91 (4.51) for the control group. The groups did not differ statistically in the above measures and the standard comprehensive rehabilitation therapy (Table 1).

**Table 1.** Median values of subject's groups, interquartile range, and the value of the Mann–Whitney U test comparing groups of subjects.

| | Main Group (n = 7) | Control Group (n = 11) | *p*-Value |
| --- | --- | --- | --- |
| Age, years | 62 (10.5) | 60 (19.5) | 0.492 |
| Female, % | 28.6 | 27.3 | 1.000 |
| Higher education, % | 71.4 | 27.3 | 0.074 |
| Disease duration, month | 1.9 (0.48) | 1.1 (1.48) | 0.132 |
| Lesion volume, cm$^3$ | 2.14 (17.08) | 1.91 (4.51) | 0.809 |
| Time between first and second fMRI | 13 (2.25) | 10 (3.75) | 0.261 |

The study was conducted in accordance with the principles of biomedical ethics formulated in the 1964 Helsinki Declaration and its subsequent updates, as well as approved by the local bioethical committees of the Federal Research and Clinical Center of Intensive Care Medicine and Rehabilitology and the Institute of Higher Nervous Activity and Neurophysiology of the RAS (Moscow). Each study participant submitted voluntary written informed consent signed by him or her after explaining the potential risks and benefits as well as the nature of the study.

### 2.2. Neuropsychological Assessment

We conducted a set of neuropsychological examinations to assess the cognitive functions of the patients: the Frontal Assessment Battery (FAB) to screen for dementia with predominantly frontal lobes or subcortical structures, and the Montreal Cognitive Assessment (MoCa) scale as a screening tool for mild and moderate cognitive impairment, including the diagnosis of poststroke conditions [16,17]. The MoCa consists of 11 tasks to assess the preservation of visual–spatial skills, executive functions, abstract thought, spatial and temporal orientation, verbal skills, attention, and memory. We additionally investigated sensorimotor reaction time and attention span using the Schulte table, which includes 4 trials. The Schulte test consists of matrices of 5 × 5 randomly arranged numbers from 1 to 25. In this assessment, subjects performed a sequential search for numbers (from the smallest number (1) to the largest (25)), simultaneously indicating and calling the retrieved numbers. We assessed the task completion time for each test separately and calculated a performance parameter (average time to find numbers over the test completion time). Auditory–verbal memory capacity was investigated using Luria Memory Words Test, considering the noted working memory impairment [18]. This test consists of 10 words, read aloud with a one-second interval between each word. We conducted 4 trials of each short-term memory task. Each trial is followed by a free recall answer and by a free estimation of recall. The words are presented in a fixed order throughout the trial. Instructions are repeated before each trial to avoid their being forgotten. After a half-hour break, we evaluated long-term memory in one trial.

The primary neuropsychological examination of the patients included in the study revealed mild to moderate cognitive impairments, as well as mild to moderate impairments

of control functions (only three patients showed normal scores on the FAB test). All patients revealed impairments in the background (neurodynamic) component of cognitive functions to a moderate to mild degree.

## 2.3. BCI Training Protocol

EEG was recorded monopolar in eight leads: P3, Pz, P4, Po7, Po8, O1, Oz, and O2. We used the electrode on the right earlobe as a reference and the electrode at position Fp1 as the middle point of the amplifier ("ground").

During the session, subjects sat at a table with a monitor displaying visual (grapheme) stimuli. The stimuli were presented in the BCI-P300 paradigm and were organized as a matrix of nine rows and five columns, with cells containing letters of the Russian alphabet. Stimulus environment and algorithms of the hardware–software complex "NeuroChat" were used for the study. Each session consisted of two blocks: the classifier training phase and the typing phase. Patients were asked to respond to the target cell lights in a way that was convenient for them: for example, by saying a short word to themselves ("once", "yes", "hurrah", etc.) or by naming the target letter or by moving the finger of a nonparetic limb. When the classifier learning phase was successfully completed, the subjects were asked to move on to the typing phase of the target text. In this phase, the task of the subject was to enter certain words on the screen using the commands of the BCI: frequent nouns of 3–5 letters in the nominative case, singular or plural. To select a letter, the subject had to follow the same instructions as in the classifier training, i.e., to successively focus on the highlights of the desired letter until it appeared in the typing line. The cell with the letter recognized by the classifier was illuminated, and the corresponding letter appeared in the upper line of the set. After 5 s, a new stimulation cycle was started to select the next letter. Incorrectly entered letters were not erased; in the case of an error, the patient proceeded to the next letter in the word. A card with a large-printed word was placed next to the screen. Patients were asked to type at least three words per session, and if there was time and motivation, six words. If the patient was very tired or time-constrained, the session was terminated regardless of the number of words typed. New words were suggested for each attempt. At the patient's request, there was a short pause for a rest between word attempts.

After successfully entering the set of six words, some patients expressed a desire to use free typing and typed their own text with the help of the BCI, but the results of such typing were not taken into account. In rare cases, at the patient's request, the task was made more difficult, and the number of words offered was increased to nine words per session.

In the course of this kind of task, the patient has to perform a number of functions: (1) hold attention on the target cell, (2) be alert and in a state of expectation, (3) hold the target letter in memory, (4) suppress distractors, (5) detect errors, (6) be rewarded with a typed letter and a positive emotional reaction from the assistant.

The target duration of the training course was 10 days. This duration was determined by the patient's average hospital stay (14 days). About 4 days were spent on primary and repeated fMRI and neuropsychological examination, and we tried to use the remaining 10 days to the maximum extent possible for training with the BCI. Our previous observations suggest that this particular duration, on the one hand, does not tire the patients and, on the other hand, does not bore them [15].

Each patient underwent from 5 to 11 sessions of BCI training. The number of sessions varied depending on the patient's condition and the duration of their treatment. Sessions were conducted every working day.

## 2.4. Resting-State fMRI Data Acquisition, Preprocessing, and Connectivity Analysis

Functional and anatomical images were acquired on a 1.5T Siemens Essenza scanner with an 8-channel head coil. Each resting-state functional run consisted of 300 T2*-weighted echoplanar images (EPIs) ($3.9 \times 3.9$ mm in-plane voxel size, 35 4.0 mm slices, interslice gap 0.8 mm, repetition time (TR) = 3670 ms, echo time (TE) = 70 ms, $64 \times 64$ matrix). In addition to the functional images, we collected a high-resolution T1-weighted anatomical scan for each

participant (192 slices, resolution $1 \times 1 \times 1$ mm, TR = 10 s, TE = 4.76 ms, $256 \times 256$ acquisition matrix).

Patients were instructed to relax, lie still with their eyes closed, and not to think about anything in particular. Patients were not sedated during the scanning procedures before and after training.

The data were processed using the CONN functional connectivity toolbox, version 19c, and SPM12. First 2 dummy volumes were excluded from analysis. The preprocessing procedure consisted of: manual AC–PC reorienting, realignment of functional images (motion correction), slice timing correction, coregistration, segmentation of structural data, normalization into standard stereotactic Montreal Neurological Institute (MNI) space, outlier detection/scrubbing using the artifact detection tool (ART) (http://www.nitrc.org/projects/artifact_detect, accessed on 1 March 2021), and spatial smoothing with a Gaussian kernel of 8 mm full width at half maximum.

Denoising was performed by removing the following confounders by linear regression: the blood-oxygen-level dependent (BOLD) signal from the white matter and CSF masks (5 principal components of each signal), scrubbing (the number of regressors corresponded to the number of identified invalid scans), and motion regression (12 regressors: 6 motion parameters, 16 first-order temporal derivatives). In addition, manually created lesion masks in the MRIcron toolbox were included in the denoising step to regress out any lesion-related signal. The resulting signals were band-pass filtered in the range of 0.008–0.12 Hz.

To test each of our three hypotheses, we used a specific set of regions of interest. We used the WFU PickAtlas (RRID:SCR_007378) [19] to construct regions of interest not contained in the CONN software package's atlas. We used anatomical structures associated with the generation of the P300 potential, according to the literature, as seeds to perform a seed-based analysis of functional connectivity. These areas include the superior frontal gyrus, middle frontal gyrus, inferior frontal gyrus, anterior cingulate cortex, precentral cortex, parietal cortex, posterior cingulate cortex, occipitotemporal cortex, cuneus, thalamus, and cerebellum [20–26]. We performed a seed-based functional connectivity analysis using a mixed-design analysis of variance (RM-ANOVA $2 \times 2$) to investigate the main effects of "group" (2 levels: BCI trained, control) and "session" (2 levels: pre, post) and their interaction. All results are presented using a statistical significance threshold at the cluster level of $p < 0.05$ and adjusting for multiple comparisons (FWE).

## 3. Results

### 3.1. Neuropsychological Assessment

The primary neuropsychological examination of the patients included in the study revealed mild to moderate cognitive impairments, as well as mild to moderate impairments of control functions (only three patients showed normal scores on the FAB test). All patients revealed impairments of the background (neurodynamic) component of cognitive functions to a moderate to mild degree. A comparison of patients before and after training showed no statistically significant changes compared to the control group.

### 3.2. Seed-Based Functional Connectivity Analysis

Our analysis revealed significantly increased functional connectivity of the right thalamic area in the main group after BCI training compared with controls. An increase in right thalamic functional connectivity was observed in three anatomical areas: the left cerebellum ($F_{2,15}$ = 15.10, $p < 0.05$, FWE-corrected), right fusiform gyrus (occipital part), and right calcarine cortex ($F_{2,15}$ = 17.43, $p < 0.05$, FWE-corrected), effect size (f) = 0.47, power (1-β) = 0.62 (Table 2).

**Table 2.** Resting-state functional connectivity, RM-ANOVA 2 × 2.

| Seed | Region | MNI (x, y, z) | Cluster Size | $F_{2,15}$ | p-FWE |
|---|---|---|---|---|---|
| R thalamus | L cerebellum | −14 −64 −18 | 287 | 17.4 | 0.000 |
| | R fusiform gyrus (occipital part) | 26 −80 −18 | | | |
| | R calcarine cortex | 6 −84 6 | | | |
| | L cerebellum | −22 −48 −46 | 39 | 15.1 | 0.027 |
| Superior frontal gyrus | | | | | |
| Middle frontal gyrus | | | | | |
| Inferior frontal gyrus | | | | | |
| Anterior cingulate cortex | | | | | |
| Precentral cortex | No significant clusters | | | | |
| Parietal cortex | | | | | |
| Posterior cingulate cortex | | | | | |
| Occipitotemporal cortex | | | | | |
| Cuneus | | | | | |
| Cerebellum | | | | | |

Figure 2 shows the results of comparing the mean values of the functional connectivity of the right thalamus before (shown in blue) and after (shown in red) a 10-day training course of patients in the BCI environment. Post hoc pairwise comparisons showed a significant increase in functional connectivity in the right thalamus to the right cerebellum (T = 10.74, $p < 0.05$, FWE-corrected) (in the figure on the left) and right superior frontal gyrus (prefrontal cortex) (T = 8.75, $p < 0.05$, FWE-corrected) after BCI training (in the figure on the right) (Figure 2).

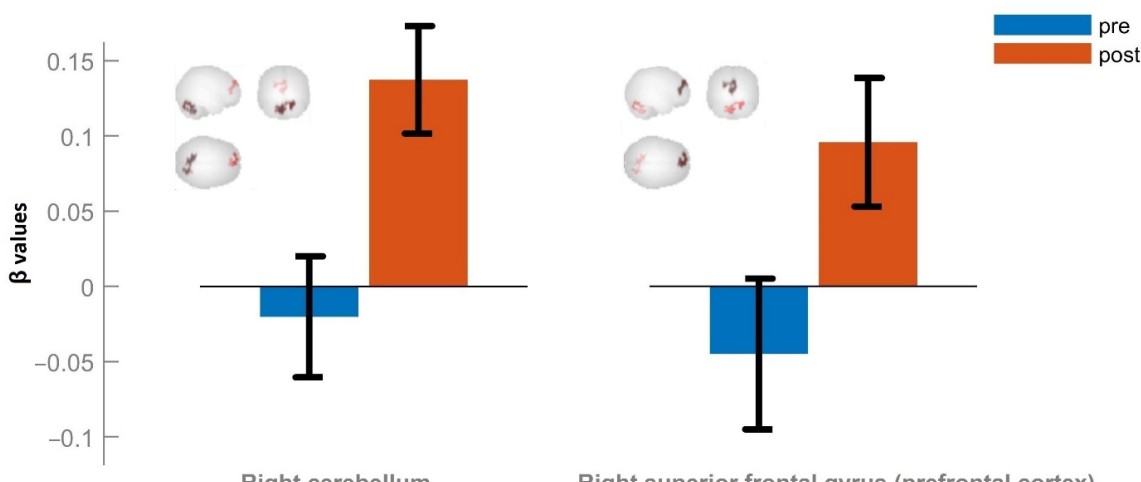

**Figure 2.** Significant clusters from the right thalamus seed in the main group (T > 8.75, p-FWE < 0.05). β values correspond to Fisher-transformed correlation coefficient values. Error bars denote standard error of the mean.

For the control group, the results of the posterior pairwise comparison are shown in Figure 3. Functional connectivity of the right thalamus at the first control point is shown in blue; functional connectivity of the thalamus after 10 days of a standard rehabilitation hospital stay is shown in red.

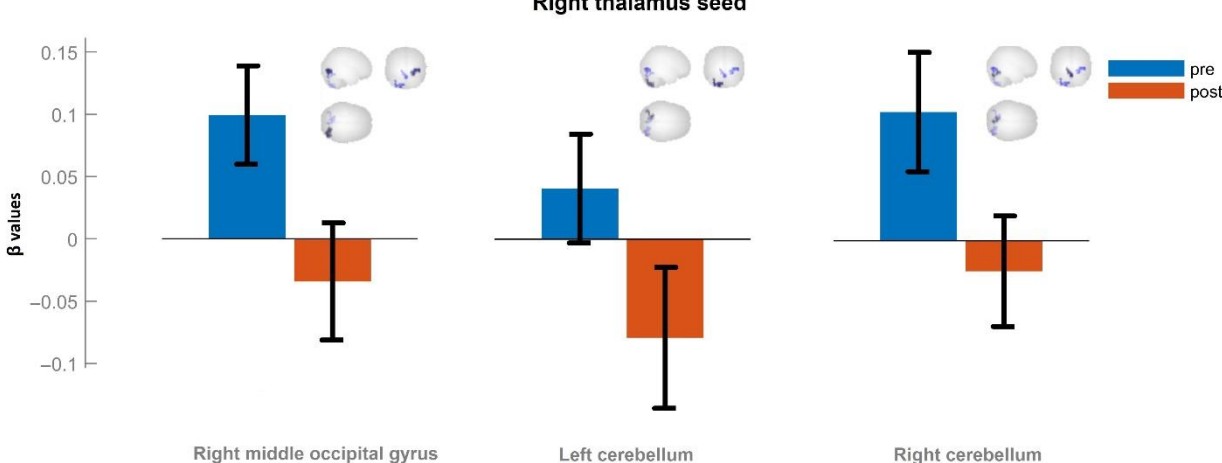

**Figure 3.** Significant clusters from the right thalamus seed in the control group (T > 6.96, p-FWE < 0.05). β values correspond to Fisher-transformed correlation coefficient values. Error bars denote standard error of the mean.

There was a significant decrease in functional connectivity in the right thalamus to the right middle occipital gyrus (T = 8.54, $p < 0.05$, FWE-corrected) (the left-hand pair of columns in the figure), left cerebellum (T = 7.52, $p < 0.05$, FWE-corrected) (the central pair of columns in the figure), and right cerebellum (T = 6.96, $p < 0.05$, FWE-corrected) (the right-hand pair of columns in the figure) (Figure 3).

Figures 2 and 3 also schematically show the localizations of the corresponding brain regions.

We did not find any increase in thalamic or other ROIs connectivity in this group.

Thus, the main effect was due to both an increase in functional connectivity between the thalamus and cerebellum in the main group and a decrease in functional connectivity between these areas in the control group.

## 4. Discussion

In the present work, we conducted a controlled pilot study of reorganization in functional brain connectivity after a 10-day BCI training course in poststroke patients. The areas of the brain that make up the neuronal circuitry of the P300 evoked potential were examined. The effect was obtained for the functional connectivity of the right thalamic region.

It is now known that the thalamus, besides the processing of sensory information, is involved in the realization of several higher cognitive functions. According to clinical studies, damage to the anterior thalamus leads to impaired memory [27], and damage to the middle thalamus leads to impaired attention and other executive functions [28]. Neuroimaging studies have shown that the thalamus is the key subcortical node of the central executive network, which supports working memory, flexibility, and inhibition [29].

Thalamic areas have also been described as part of the cerebello–thalamocortical pathway, which is also involved in implementing executive functions [30]. In this regard, the enhancement of thalamic–cerebellar connectivity in the main group after BCI training compared with the control group seems to be an encouraging result. The increased connectivity of the right thalamus to the primary (right calcarine cortex) and secondary (occipital part of the fusiform gyrus) visual cortex may reflect processes of activating grapheme processing circuits (taking into account the nature of our BCI paradigm). The left posterior fusiform gyrus is thought to be responsible for grapheme and word processing, but in patients

with lesions of the left fusiform gyrus, word processing also shows activation in the right fusiform gyrus [31].

Of particular note are the results from a pairwise comparison of the functional connectivity data in the main group. We obtained increased connectivity between the thalamus and the prefrontal cortex and cerebellum. It seems intriguing that all these areas are part of the cerebello–thalamocortical pathway involved in, as mentioned above, executive functions [30] and that the prefrontal cortex is associated with the domain of executive functions, while the thalamic–prefrontal cortex connection (especially the dorsolateral prefrontal cortex) is implicated in attentional processing [32,33].

The decreased functional connectivity in the control group between the thalamus and cerebellum (both right and left) in the group of delayed therapy is probably due to a lack of activation in the neuronal circuitry in the absence of specific training in the form of prolonged concentration and retention of the target symbol in memory.

## 5. Limitations

Although our results indicate a modulating effect of the technique on the intrinsic functional connectivity of the brain, the weakness of our study is the small number of subjects, which increases the likelihood of a nonrepresentative sample. The cross-sectional pattern of change in mean values of functional connectivity at the first and second study points, as well as differences in mean values at the first control point, confirm our fears. It should also be noted that the statistical power of our study, considering the sample size, was 0.62. This value corresponds to a very high probability of a type II error. Thus, the study calls for a further increase in sampling. The effect size obtained empirically in our study was 0.47; accordingly, the required sample size to achieve a statistical power of 0.9 is 28 participants [34].

In the main group, there was a higher proportion of those who received a higher education. This raises the question of whether the effect obtained in our study is a consequence of such a disproportion, i.e., a consequence of a higher level of education in the main group. The level of education in the poststroke disorder clinic is part of a pool of indicators of cognitive reserve, which provides more effective resistance to the damaging effects of stroke on the brain and cognitive function [35]. Cognitive reserve may also influence stroke rehabilitation [36,37]. Regarding our study, it is highly unlikely that the presence of higher levels of education alone can enhance the functional connectivity of neuronal circuits associated with implementing control functions after stroke. The short period (10 days) during which these changes occurred is itself evidence against the influence of cognitive reserve on this process. The fact is that the influence of cognitive reserve has underlying "compensatory" (slow) rather than "spontaneous" (fast) mechanisms [37]. The absence of a difference in neuropsychological examination between the main group and the control group further reduces the probability that this factor plays a dominant role. Most likely, such a disproportion may have had only a modulating effect in reinforcing the observed effect. On the other hand, if education level had a significant effect, the differences we obtained between the groups should have been evident in both the secondary fMRI session and the primary fMRI session. However, since our results showed significant differences between the groups only for the secondary session, we attribute these differences to the effectiveness of BCI training, but not to education level. Nevertheless, in our future study, we will do our best to avoid such a disproportion, not to say that the question of its occurrence is itself subject to investigation.

We also did not perform parcellation of the thalamus to the nuclei specific for cognitive functions, which is also a drawback of our study.

Our future work in this study will focus on increasing statistical power by increasing sample size and homogeneity and improving the anatomical accuracy of neuroimaging data and analysis.

## 6. Conclusions

Our preliminary results confirm the existence of a modulating effect on brain functional connectivity as a result of 10 days of training in a P300-based BCI speller environment.

This effect is realized by enhancing functional integration in the neuronal circuits involved in modulating cognitive processes (cerebello–thalamocortical pathway).

Changes in functional connectivity of the thalamus at rest may represent a functional substrate of cognitive improvement/reduction in the presence/absence of intensive cognitive training using the P300-based BCI speller in patients after stroke.

**Author Contributions:** Conceptualization, L.M.; methodology, L.M. and A.K.; formal analysis, L.M. and A.K.; investigation, M.R., V.Z., V.S. and P.P.; data curation, L.M. and M.R.; writing—original draft preparation, L.M.; writing—review and editing, L.M. and M.P.; supervision, A.G. All authors revised the article. All authors have read and agreed to the published version of the manuscript.

**Funding:** This research received no external funding.

**Institutional Review Board Statement:** The study was conducted in accordance with the principles of biomedical ethics formulated in the 1964 Helsinki Declaration and its subsequent updates and was approved by the local bioethical committee of the Federal Research and Clinical Center of Intensive Care Medicine and Rehabilitology (Moscow) (Protocol No. 1/20/2). All participants submitted voluntary written informed consent signed by him or her for the publication of any potentially identifiable images or data included in this article.

**Informed Consent Statement:** Informed consent was obtained from all subjects involved in the study.

**Data Availability Statement:** The raw data supporting the conclusions of this article will be made available by the authors without undue reservation.

**Conflicts of Interest:** The authors declare that the research was conducted in the absence of any commercial or financial relationships that could be construed as a potential conflict of interest.

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
