# Peer review of "Rapid Effects of BCI-Based Attention Training on Functional Brain Connectivity in Poststroke Patients: A Pilot Resting-State fMRI Study"

_2035-8377, doi:10.3390/neurolint15020033_

Round 1

Reviewer 1 Report (Previous Reviewer 1)

The paper can be accepted in its current form

Reviewer 2 Report (Previous Reviewer 2)

This manuscript is a resubmission of a previous version. The authors have made significant efforts to address the concerns of the reviewer. In particular, I found the revised figures (Fig.2 and 3) to be much improved. As a result, the paper is much clearer and presents the new insights of BCI-based training in ischemic stroke patients. Thus, I recommend publication without re-review.

This manuscript is a resubmission of an earlier submission. The following is a list of the peer review reports and author responses from that submission.

Round 1

Reviewer 1 Report

Thank you for inviting me to review this manuscript.
In this study the authors attempted to evaluate the modulation of neuronal circuits under the influence of 10 days of training in a P3-based BCI-speller in subacute ischemic stroke patients.
Statistical analyses have well conducted and figures and tables are informative. 
The paper is interesting but I have some concerns:
1)The smallness of the sample size could be a real problem for this type of paper. The authors could test the power of their sample size to strengthen their study.
2) I have noticed important differences between groups in % of higher education. The authors should comment that this aspect cannot influence the results.

Author Response

Response to Reviewer 1 Comments

Point 1: The smallness of the sample size could be a real problem for this type of paper. The authors could test the power of their sample size to strengthen their study.

Response 1: Dear reviewer, thank you for your comments, we calculated the effect size and statistical power of the study, they turned out to be equal: effect size (f)=0.47, power (1-β)=0.62 and are given on line 258 of the manuscript. We then provide the following text in the section discussing the limitations of the study: " We should also be noted that the statistical power of our study, considering the sample size, was 0.62. This value corresponds to a very high probability of a type II error. Thus, the study calls for a further increase in sampling. The effect size obtained empirically in our study was 0.47; accordingly, the required sample size to achieve a statistical power of 0.9 is 28 participants [34].» in lines 330-334.

Point 2: I have noticed important differences between groups in % of higher education. The authors should comment that this aspect cannot influence the results.

Response 2: Thank you for the helpful observation, in accordance with the correction we bring the discussion:" In the main group there was a higher proportion of those who received a higher education. This raises the question whether the effect got in our study is not a conse-quence of such a disproportion, i.e. a consequence of a higher level of education in the main group? The level of education in the post-stroke disorder clinic is part of a pool of indicators of cognitive reserve, which provides more effective resistance to the damag-ing effects of stroke on the brain and cognitive function [35]. Cognitive reserve may al-so influence stroke rehabilitation [36,37]. Regarding our study, it is highly unlikely that the presence of higher levels of education alone can enhance the functional connectivi-ty of neuronal circuits associated with implementing control functions after stroke. The short period (10 days) during which these changes occurred is itself evidence against the influence of cognitive reserve on this process. The fact is that the influence of cognitive reserve has underlying "compensatory" (slow) rather than "spontaneous" (fast) mechanisms [37]. The absence of a difference in neuropsychological examination between the main group and the control group further reduces the probability that this factor plays a dominant role. Most likely, such a disproportion may have had only a modulating effect in reinforcing the observed effect. On the other hand if education level had a significant effect, the differences we obtained between the groups should have been evident in both the secondary fMRI session and the primary fMRI session. However, since our results showed significant differences between the groups only for the secondary session, we attribute these differences to the effectiveness of BCI train-ing, but not to the education level.  Nevertheless, in our future study, we will do our best to avoid such a disproportion, not to say that the question of its occurrence is itself subject to investigation.» in lines 335-356.

Reviewer 2 Report

In the present manuscript entitled “Rapid effects of BCI-based attention training on functional brain connectivity in post-stroke patients: a pilot resting-state fMRI study”, Mayorova et al. presented that 10-days-training in a BCI-speller leads to functional improvement, increased connectivity between thalamus and the prefrontal cortex and the cerebellum. The aftereffects of stroke are a major problem in modern society because they greatly affect quality of life. The plan is well thought out. Additionally, the interpretation of the results as well as consideration of limitations of studies, are well described. It is very interesting to identify that the thalamus is involved in the expression of cognitive functions. Following are few issues for this study.

1) I think it would be better to plot each value in the Figures, 2 and 3.

2) Please mention how you decided to train 10 days long?

3) The main group tends to have a higher proportion of those who have received higher education. Will this affect the protocol?

Minor point:

4) It would be better to add the description of the illustration inserted in the Figures, 2 and 3.

Author Response

Response to Reviewer 2 Comments

Point 1: I think it would be better to plot each value in the Figures, 2 and 3.

Response 1: Dear reviewer, thank you for your comments, unfortunately, the program for processing functional connectivity does not allow for the construction of figures with the designation of values for each participant. After many and fruitless attempts we decided to leave this issue to your discretion and the editor's discretion.

Point 2: Please mention how you decided to train 10 days long?

Response 2: Thank you, we indicated what guided us in lines 197-202 of the manuscript " The target duration of the training course was 10 days. This duration was deter-mined by the patient's average hospital stay (14 days). About 4 days were spent on primary and repeated fMRI and neuropsychological examination, and we tried to use the remaining 10 days to the maximum extent possible for training with the BCI. Our previous observations suggest that this particular duration, on the one hand, does not tire the patients, and on the other hand, does not bore them [15].»

Point 3: The main group tends to have a higher proportion of those who have received higher education. Will this affect the protocol?

Response 3: Thank you for the helpful observation, in accordance with the correction we bring the discussion:" In the main group there was a higher proportion of those who received a higher education. This raises the question whether the effect got in our study is not a conse-quence of such a disproportion, i.e. a consequence of a higher level of education in the main group? The level of education in the post-stroke disorder clinic is part of a pool of indicators of cognitive reserve, which provides more effective resistance to the damag-ing effects of stroke on the brain and cognitive function [35]. Cognitive reserve may al-so influence stroke rehabilitation [36,37]. Regarding our study, it is highly unlikely that the presence of higher levels of education alone can enhance the functional connectivi-ty of neuronal circuits associated with implementing control functions after stroke. The short period (10 days) during which these changes occurred is itself evidence against the influence of cognitive reserve on this process. The fact is that the influence of cognitive reserve has underlying "compensatory" (slow) rather than "spontaneous" (fast) mechanisms [37]. The absence of a difference in neuropsychological examination between the main group and the control group further reduces the probability that this factor plays a dominant role. Most likely, such a disproportion may have had only a modulating effect in reinforcing the observed effect. On the other hand if education level had a significant effect, the differences we obtained between the groups should have been evident in both the secondary fMRI session and the primary fMRI session. However, since our results showed significant differences between the groups only for the secondary session, we attribute these differences to the effectiveness of BCI train-ing, but not to the education level.  Nevertheless, in our future study, we will do our best to avoid such a disproportion, not to say that the question of its occurrence is itself subject to investigation.» in lines 335-356.

Point 4: It would be better to add the description of the illustration inserted in the Figures, 2 and 3.

Response 4: We have added a description of figures 2 and 3 in lines 261-267 and 272-282 of the manuscript " Figure 2 shows the results of comparing the mean values of the functional connec-tivity of the right thalamus before (shown in blue) and after (shown in red) a 10-day training course of patients in the BCI environment. Post-hoc pairwise comparisons showed a significant increase in functional connectivity in the right thalamus to the right cerebellum (T=10.74, p<0.05, FWE-corrected) (in the figure on the left) and right superior frontal gyrus (prefrontal cortex) (T=8.75, p<0.05, FWE-corrected) after BCI-training (in the figure on the right) (Figure 2). … For the control group, the results of the posterior pair-wise comparison are shown in Figure 3. Functional connectivity of the right thalamus at the first control point is shown in blue; functional connectivity of the thalamus after 10 days of a standard rehabilitation hospital stay is shown in red.

There was a significant decrease in functional connectivity in the right thalamus to the right middle occipital gyrus (T=8.54, p<0.05, FWE-corrected) (the left-hand pair of columns in the figure), left cerebellum (T=7.52, p<0.05, FWE-corrected) (the central pair of columns in the figure), and right cerebellum (T=6.96, p<0.05, FWE-corrected) (the right-hand pair of columns in the figure) (Figure 3).

Figures 2 and 3 also show schematically the localizations of the corres
